# Mesothelial Cells Exhibit Characteristics of Perivascular Cells in an In Vitro Angiogenesis Assay

**DOI:** 10.3390/cells12202436

**Published:** 2023-10-11

**Authors:** Chrysa Koukorava, Kelly Ward, Katie Ahmed, Shrouq Almaghrabi, Sumaya Dauleh, Sofia M. Pereira, Arthur Taylor, Malcolm Haddrick, Michael J. Cross, Bettina Wilm

**Affiliations:** 1Department of Pharmacology and Therapeutics, Institute of Systems, Molecular and Integrative Biology, University of Liverpool, Liverpool L69 3GE, UK; 2Department of Molecular Physiology and Cell Signalling, Institute of Systems, Molecular and Integrative Biology, University of Liverpool, Liverpool L69 3GE, UK; 3Medicines Discovery Catapult, Alderley Park, Macclesfield SK10 4ZF, UK; 4Department of Women’s and Children’s Health, Institute of Life Course and Medical Sciences, University of Liverpool, Liverpool L69 3GE, UK

**Keywords:** perivascular cells, mesothelial cells, in vitro angiogenesis assay, cell shape, cell migration, mesothelial-mesenchymal transition, Ng2, Zeb1, cardiac spheroids

## Abstract

Mesothelial cells have been shown to have remarkable plasticity towards mesenchymal cell types during development and in disease situations. Here, we have characterized the potential of mesothelial cells to undergo changes toward perivascular cells using an in vitro angiogenesis assay. We demonstrate that GFP-labeled mesothelial cells (GFP-MCs) aligned closely and specifically with endothelial networks formed when human dermal microvascular endothelial cells (HDMECs) were cultured in the presence of VEGF-A_165_ on normal human dermal fibroblasts (NHDFs) for a 7-day period. The co-culture with GFP-MCs had a positive effect on branch point formation indicating that the cells supported endothelial tube formation. We interrogated the molecular response of the GFP-MCs to the angiogenic co-culture by qRT-PCR and found that the pericyte marker *Ng2* was upregulated when the cells were co-cultured with HDMECs on NHDFs, indicating a change towards a perivascular phenotype. When GFP-MCs were cultured on the NHDF feeder layer, they upregulated the epithelial–mesenchymal transition marker *Zeb1* and lost their circularity while increasing their size, indicating a change to a more migratory cell type. We analyzed the pericyte-like behavior of the GFP-MCs in a 3D cardiac microtissue (spheroid) with cardiomyocytes, cardiac fibroblasts and cardiac endothelial cells where the mesothelial cells showed alignment with the endothelial cells. These results indicate that mesothelial cells have the potential to adopt a perivascular phenotype and associate with endothelial cells to potentially support angiogenesis.

## 1. Introduction

The peritoneum is lined by the mesothelium, a monolayer of predominantly squamous-like epithelial cells, which secrete lubricants, allowing friction-free organ movement, and maintain homeostasis within the cavity. The mesothelium arises from the mesoderm and is also found in the pleural and pericardial serosal cavities; in all cavities, the mesothelium plays an important role in tissue homeostasis and the immune response by secreting inflammatory mediators, growth factors and extracellular matrix components [1,2,3]. Previous genetic lineage tracing studies by ourselves and others showed that mesothelial-derived cells contribute to the vasculature of the heart, lung and intestine via epithelial-to-mesenchymal transition (EMT) and differentiation into vascular smooth muscle cells [4,5,6]. However, the molecular mechanisms that regulate the contribution of mesothelial cells (MCs) to the vasculature have not been described.

Endothelial cells (ECs) line the innermost layer of vascular blood vessels (arteries, veins and capillaries) and lymphatic vessels and play critical roles in blood flow, permeability and general homeostasis, as well as immune/inflammatory responses [7,8]. ECs also originate from the mesoderm and are formed during vasculogenesis [9,10]. Vasculogenesis and angiogenesis are the fundamental processes by which new blood vessels are formed. Vasculogenesis is defined as the differentiation of endothelial precursor cells, or angioblasts, into endothelial cells and the de novo formation of a primitive vascular network [11]. By contrast, during angiogenesis, blood vessels emerge from pre-existing vessels. Angiogenesis is a key mechanism in embryonic development and the pathogenesis of diseases such as cancer, inflammation and diabetic retinopathy [12,13].

Angiogenesis is a complex, highly regulated process comprising a multifactorial cascade of events that involves the degradation of components of the extracellular matrix, followed by the migration, proliferation and differentiation of endothelial cells to form lumen-containing vessels [14,15]. Vessel maturation also requires the coordinated recruitment of mural cells such as pericytes, which regulate endothelial cell proliferation, differentiation and permeability, all of which are essential for the pathophysiological function of blood vessels [16,17].

In order to analyze the interactions of MCs in angiogenic processes, we turned to a modified in vitro angiogenesis assay of primary human fibroblasts and primary human microvascular endothelial cells [18,19,20,21,22,23] in which stable endothelial tubes form within 4–7 days. We used this system to interrogate the interaction of MCs [24], with the human ECs in the in vitro assay.

Our data reveal that MCs have the ability to intimately associate with ECs when plated in a 2D monolayer or 3D multicellular cardiac spheroids. The perivascular nature of the MCs in close proximity to ECs was confirmed by gene expression analysis which revealed upregulation of the pericyte marker *Ng2*. Furthermore, analysis of MC morphology revealed that contact with fibroblasts facilitated a change in MC shape indicative of a migratory phenotype. Gene expression analysis also revealed that the association of the MCs with fibroblasts was the critical event in facilitating upregulation of *Ng2* and expression of the EMT marker *Zeb1*. Our data suggest that MCs have the ability to acquire a perivascular phenotype and associate with endothelial cells to potentially support angiogenesis.

## 2. Materials and Methods

All reagents were purchased from Sigma (Poole, UK) unless otherwise stated. All cells were cultured at 37 °C and 5% CO_2_ in a humidified incubator.

### 2.1. Basic Cell Culture

Normal human dermal fibroblasts (NHDFs, C-12300), human dermal microvascular endothelial cells (HDMECs, C-12210), human cardiac microvascular endothelial cells (HCMECs, C-12285), human dermal lymphatic endothelial cells (HDLECs, C-12216), human cardiac fibroblasts (HCFs, C-12375) and human pericytes (PCs, C-12980) were purchased from PromoCell (Heidelberg, Germany), human keratinocyte cells (HaCaTs) [25] were a gift from Dr A Simpson (University of Liverpool), the human cardiomyocyte cell line AC16 (#SCC109) was purchased from Millipore and HEK 293TN (LV900A-1) cells were purchased from System Biosciences (Palo Alto, CA, USA). Mouse mesothelial cells (MCs) were isolated as described [24] and cultured in Dulbecco’s Modified Eagle’s Medium (DMEM; D5796) supplemented with 10% fetal bovine serum (FBS; F6178) (MC medium), HaCaTs in DMEM (D6429) supplemented with 10% FBS and 1% Penicillin/Streptomycin (P/S) and HEK 293 cells in DMEM (D6546) supplemented with 10% FBS, 200 mM L-glutamine (G7513) and 1% Penicillin/Streptavidin on untreated tissue culture plastic. HDMECs and HCMECs were routinely cultured in Endothelial Cell Growth Medium MV 2 (EGM MV2; C-22022) with supplement mix (C-39226, all PromoCell) and NHDFs and HCFs in Fibroblast Growth Medium (FGM; C-23010) with supplement mix (C-39315, all PromoCell) on tissue culture plastic coated with 0.5% (*w*/*v*) gelatin. Basal Endothelial Cell Growth Medium (C-22022, PromoCell, Heidelberg, Germany) supplemented with 1% FBS (bMV2) was used during the angiogenesis assays. Human endothelial cells were gently dissociated and released from the culture dishes using accutase (C-41310, Promocell, Heidelberg, Germany). The vascular endothelial authenticity of HDMEC and HCMEC batches used in this study was determined by qRT-PCR analysis to confirm minimal contamination with lymphatic endothelial cells (Appendix A) and included detection of the lymphatic markers *VEGFR3/FLT4*, *PODOPLANIN* and *PROX1* (Appendix A) [26,27,28,29].

CD1 Mouse primary Artery Smooth Muscle Cells (MASMC) #CD-1081 were grown in smooth muscle cell media #M2268. CD1 Mouse primary Dermal Microvascular Endothelial Cells (MDMEC) #CD1064 and C57BL/6 Mouse primary Cardiac Microvascular Endothelial Cells (MCMEC) were routinely cultured in endothelial cell media #M1168. All primary mouse cells and associated media were purchased from Cell Biologics (Chicago, IL, USA).

### 2.2. Lentiviral Infections

The producer cell line HEK 293TN was transfected with the packaging (psPAX2, Addgene 12260), envelope (pMD2.G, Addgene 12259) and transfer (pLNT-SFFV-GFP [24] or pHIV-dTomato, Addgene 21374) plasmids via the calcium-phosphate method (CAPHOS, Sigma). Supernatant was collected and either used in its crude form to transduce cells or concentrated via ultracentrifugation. Target cells PCs, HaCaTs (GFP), or HDMECs (dTomato) were transduced at ~60% confluence while in medium containing polybrene (8 µg/mL) using the GFP or dTomato (dT) lentivirus. Transduced cells GFP-PC, GFP-HaCaT and dT-HDMEC were expanded and sorted via fluorescence-activated cell sorting (FACS) analysis. GFP-expressing MCs (GFP-MCs) had been generated as described [24].

### 2.3. Flow Cytometry

Flow cytometry was performed on a FACSCalibur sorter (BD Biosciences, Wokingham, UK) using the 488 nm laser and the FL1 detector to detect the GFP transgene and the FL2 detector to detect the dTomato transgene. Forward- and side-scatter characteristics determined the exclusion of dead cells. Untransduced cells were used as a control. FACS data were analyzed using Cyflogic Software version 1.2.1 (Cyflo, Turku, Finland).

### 2.4. Cell Culture Experiments

#### 2.4.1. GFP-MC Stimulation

A total of 3 × 10^4^/well GFP-MCs were seeded in a 24-well plate in full-growth MC medium and grown for 3 days. MC medium was refreshed every other day. On day 3, cells were either given fresh MC medium or cultured in basal MC medium (1% FBS) or in basal MC medium supplemented with 50 ng/mL recombinant human VEGF-A_165_ (100-20; Peprotech, London, UK) and culture for another 3 days.

#### 2.4.2. Culture of GFP-MCs on NHDFs

NHDFs (passage 4–10) were seeded at 3 × 10^4^/well on gelatinized glass-bottomed 24-well plates (662892; Greiner Bio-One, Stonehouse, UK) in FGM and cultured for 3 days. On day 1 of the experiment, 5 × 10^3^/well GFP-MCs were seeded onto the NHDFs and cultured for 24 h in full-growth MC medium. On day 2, medium was changed to basal MC medium (1% FBS) or basal MC medium supplemented with 50 ng/mL recombinant VEGF-A_165_, and the cells co-cultured for an additional 2 days.

#### 2.4.3. Basic Angiogenesis Assay

NHDFs (passages 14–19) were seeded at 1.5 × 10^4^/well on gelatinized glass-bottomed 24-well plates in FGM and cultured for 3 days [22]. On day 1 of the experiment, 3 × 10^4^/well HDMECs or dT-HDMECs were seeded onto the NHDFs in EGM MV2, either alone or in combination with GFP-MCs, GFP-PCs or GFP-HaCaTs. Seeding ratios were 1:60, 1:30, 1:15, 1:6, 1:2 or 1:1 (GFP-MCs, GFP-PCs or GFP-HaCaTs: HDMECs) for the initial comparative studies, and typical co-cultures were performed at a ratio of 1:15. On day 2, media was replaced with bMV2 ± 50 ng/mL recombinant human VEGF-A_165_, refreshed on day 4, and cells cultured until day 7. Assays included three replicates, and three different biological mesothelial preparations were used. Preliminary studies had confirmed that MC behavior was maintained when cultured in media of the angiogenesis assay.

#### 2.4.4. Cardiac Angiogenesis Assay Using HCMECs and HCFs

HCFs were seeded at 3 × 10^4^/well on gelatinized 24-well plates in FFGM and cultured for 3 days. On day 1 of the experiments, HCMECs were seeded onto the HCFs at 3 × 10^4^ cells/well with or without GFP-MCs at 5 × 10^3^ cells/well in MV2 FGM. On day 2, media was replaced with bMV2 ± 50 ng/mL recombinant human VEGF-A_165_, refreshed on day 4, and cells cultured until day 7.

#### 2.4.5. GFP-MCs in the Cardiac Spheroid Model

Following a previously published protocol for generating cardiac microtissues, spheroids containing AC16 cells, HCMECs, and HCFs were formed with a cardiomyocyte:fibroblast:endothelial cell ratio of 4:2:1 [30]. In short, AC16 cells, HCMECs, HCFs and GFP-MCs were detached using accutase, and resuspended in 4 mL of microtissue media (1 part DMEM 10% FCS and 1 part EGM MV2) at the following densities: AC16 2850 cells/mL, HCF 1420 cells/mL, HCMEC 710 cells/mL and GFP-MC 70 cells/mL. 100 μL of the mixed cell suspension was plated into each well of a round-bottomed 96-well ULA plate (CLS3474-24EA, Corning, via Sigma, Poole, UK). The following day, an additional 100 μL of bMV2 + 50 ng/mL recombinant human VEGF-A_165_ was added per well. A total of 100 μL of media was replaced every 2 days and the spheroids remained in culture for 7 days in total.

### 2.5. Boyden Chamber Assay

GFP-MCs at passage 7 were seeded at 2 × 10^4^ cells on polyethylene terephthalate (PET) membranes with 8 µm pores (354578; Scientific Laboratory Supplies, Nottingham, UK) and cultured in bMV2 (SFM) or supplemented with conditioned medium (CM), or 0.1, 1, 10, 50, or 100 ng/mL recombinant human VEGF-A_165_, or MV2 EGM for 16 h at 37 °C, 5% CO_2_. The lower side of the inserts were fixed in quick diff fix solution (102164; Reagena, Toivala, Finland) for eight minutes followed by incubation in quick diff blue solution for four minutes to stain the nuclei of migrated cells. Each condition was performed in triplicate and for each insert 12 regions of interest were imaged using a Leica Leitz DMRB microscope with a Leica DFC450 C camera. Cells from each image were counted using ImageJ (version 1.53k) software (NIH, Bethesda, MD, USA).

### 2.6. Immunofluorescence

2D cell cultures were fixed with 4% paraformaldehyde (PFA; Sigma, Poole, UK, P6148) in phosphate-buffered saline (PBS) and permeabilized with 0.25% Triton-X-100 (Sigma, Poole, UK, 93426). After blocking for 1 h in 2% bovine serum albumin (BSA; BP9701, Fisher Scientific, Loughborough, UK) in PBS, cells were incubated with primary antibodies directed against CD31 (M0823, DAKO), GFP (ab6556, Abcam, Cambridge, UK). Secondary antibodies were anti-mouse or anti-rabbit Alexa fluor 488- or 594-coupled (Life Technologies, Paisley, UK). DAPI (D1306, Life Technologies, Paisley, UK) was used for nuclear staining. Three fields of view were randomly imaged per condition.

The 3D spheroid cultures were fixed in 2% PFA in PBS overnight followed by permeabilization with 0.5% Triton-X-100 overnight. The spheroids were blocked in 3% BSA in PBS and 0.1% Triton-X-100 for 2 h followed by incubation with primary antibodies and secondary antibodies, each in 1% BSA/PBS/0.1% Triton-X-100 overnight.

Imaging was performed using a Leica DM IRB microscope with a Leica DFC420 C camera or a Zeiss AxioObserver inverted microscope with Apotome 2.0 and processed using ImageJ/Fiji (version 1.53k) [31].

### 2.7. Live Cell Imaging of GFP-MCs in Angiogenesis Co-Culture

Co-cultures of GFP-MCs, NHDFs and dTomato-labeled HDMECs in a normal angiogenesis assay were placed in a Cell IQ v.2 (Chip-Man Technologies Ltd., Tampere, Finland) automated culture and imaging platform, and images obtained using the Cy3 and Cy5 filters, on days 2–7, initially every 8 h, and then every 6 h and 15 min.

### 2.8. Image Analysis

The alignment of GFP-labeled cells to CD31-stained endothelial tubes was quantified by generating a mask using Fiji (version 1.53k) (Appendix A). Next, the mask was expanded by 5 μm in each direction to allow detection of aligned cells within this distance and determine the percentage of aligned cells. The area occupied by endothelial tubes was measured before the 5 μm expansion. Vessel length, number of branch points and vessel area were quantified using AngioTool 0.5 [32] in 3 random fields of view per well, with triplicate wells per condition, and 2 independent experiments. We assessed cell shape (area, perimeter, circularity) using Cellpose segmentation (https://colab.research.google.com/github/MouseLand/cellpose/blob/main/notebooks/Cellpose_cell_segmentation_2D_prediction_only.ipynb#scrollTo=obeWmo9R4Fb) [33]. Images were changed into black/white to outline the GFP expression only, and flow threshold parameter was set to 0.7, while cell probability threshold was set to −4. Cellpose ‘flow’ outputs were loaded into Fiji, transformed using Glasbey on dark (LUT) and each individual cell was measured for the three cell shape parameters. Tabular outputs were quantified using GraphPad Prism 10.0.2. Between 2 and 4 images per condition from two independent experiments were analyzed.

### 2.9. qRT-PCR

Total RNA isolated from mono- and co-cultures was transcribed to cDNA using Superscript III reverse transcriptase (18080044, Life Technologies, Paisley, UK), and cDNA amplified using Power SYBR Green (4368708, Life Technologies, Paisley, UK) including optimized mouse primers (Appendix A) on a ViiA7 (ThermoFisherScientific, Waltham, MA, USA). Cycles consisted of an initial incubation at 50 °C and 95 °C for 2 and 10 min, respectively, followed by 40 cycles of 95 °C for 15 s and 60 °C for 1 min.

Primer specificity and sensitivity to detect mouse transcripts within the cDNA mix containing human and mouse transcripts were ensured by selecting primers specific for mouse cDNA according to the following work-flow: (1) mouse and human mRNA sequences for the genes of interest were compared by NCBI blastn to identify unique parts within the mouse mRNA sequence (https://blast.ncbi.nlm.nih.gov/Blast.cgi?BLAST_SPEC=blast2seq&LINK_LOC=align2seq&PAGE_TYPE=BlastSearch); (2) qRT-PCR primers were designed within the unique mouse mRNA sequences and flanking regions using NCBI primer blast (https://www.ncbi.nlm.nih.gov/tools/primer-blast/index.cgi); and (3) primer specificity and efficiency were determined by analyzing the Ct value over a range of mouse heart cDNA (0pg to 10,000 pg cDNA) with and without spiked human heart cDNA (0 pg to 10,000 pg total cDNA). Primers showing an efficiency of 95–105%, with no interference from human cDNA, were selected for downstream analysis. For each sample, the average cycle threshold (CT) value was normalized to *Gapdh* and then compared to the relevant control sample using the comparative CT (2^−ΔΔCT^) method [34]. Primer sequences for mouse transcripts are given in Appendix A. For analysis of human transcripts, the average cycle threshold (CT) value was normalized to *β-ACTIN* and then compared to the relevant control sample using the comparative CT (2^−ΔΔCT^) method [34]. Primer sequences for human transcripts are given in Appendix A.

### 2.10. Statistics and Data Analysis

Quantitative analysis was performed using Microsoft Office Excel 2016–2022, and Graphpad Prism. Cell alignment, vessel parameter data, Boyden chamber cell migration data and the qRT-PCR data were analyzed using a one-way ANOVA with Tukey’s post-hoc multiple comparisons test, or an unpaired Student’s *t*-test, where *p* < 0.05 was considered statistically significant.

Data from the Cellpose cell shape analysis were tested for normal distribution, which was rejected, and analyzed for significance using the Kruskal–Wallis test followed by Dunn’s multiple comparison tests.

## 3. Results

### 3.1. Mesothelial Cells Showed Specific Alignment with Endothelial Tubes in an In Vitro Angiogenesis Assay

Using the mesothelial marker Wt1, we have previously revealed a link between vascular development and mesothelial cells [6]. In order to interrogate the capacity of MCs to adopt perivascular characteristics, we investigated whether MCs could align to endothelial tubes in vitro. We employed a previously utilized angiogenesis assay where human dermal microvascular endothelial cells (HDMECs) formed an endothelial network when cultured on a confluent layer of normal human dermal fibroblasts (NHDFs) in the presence of VEGF-A_165_ (Appendix A) [21,23,35]. To test our hypothesis, we modified the angiogenesis assay by simultaneously seeding MCs, stably labeled with a GFP lentivirus (GFP-MCs) [24] with the HDMECs in the presence and absence of VEGF-A_165_ (Figure 1A).

After 6 days of co-culture with HDMECs (5 days with/without VEGF-A_165_), we analyzed how the GFP-MCs had interacted with the endothelial tube formation process by immunostaining for GFP and CD31 to detect MCs and HDMECs, respectively. Our results revealed that in the presence of VEGF-A_165,_ the GFP-MCs had aligned quite tightly with the endothelial tubes (Figure 1B–D). Confocal imaging revealed that the GFP-MCs aligned quite specifically with HDMEC-formed endothelial tubes in the angiogenesis cultures (Appendix A). Specifically, we observed extended cell projections and specific attachment of the GFP-MCs along and around the endothelial tubes (Figure 1E). Co-culture of GFP-MCs with HDMECs at different seeding ratios (1:60 (500 GFP-MCs:30,000 HDMECs) to 1:1 (30,000 GFP-MCs:30,000 HDMECs)), in the presence of VEGF-A_165_, demonstrated specific attachment and alignment to endothelial tubes independent of their seeding number (Appendix A). By contrast, in the absence of VEGF-A_165_, no particular alignment of the GFP-MCs to the swirls of endothelial cells was observed, independent of the seeding ratio, nor was there an effect on endothelial tube formation (Appendix A).

We compared the alignment behavior shown by the GFP-MCs with that of human-placenta-derived pericytes (PCs) in co-culture with HDMECs. As before, PCs labeled with GFP lentivirus (GFP-PCs) were seeded with the HDMECs and analyzed for their alignment after 6 days. Similar to GFP-MCs, we observed that GFP-PCs tended to be situated near the endothelial tubes, although in parallel orientation and not as closely attached as the GFP-MCs (Figure 1F–I). This arrangement of GFP-PCs towards the endothelial tubes was not changed when the pericytes were seeded at different ratios to the HDMECs (1:1 to 1:60) in the presence of VEGF-A_165_ (Appendix A).

We also tested the interaction of cells of a human keratinocyte cell line (HaCaT cells) in co-culture with HDMECs as a negative control. In the co-culture with HDMECs, HaCaT cells labeled with GFP lentivirus (GFP-HaCaTs) formed patches over and between the endothelial tubes and failed to align with the tubular cells (Figure 1J–M). This observation was independent of the seeding ratio between GFP-HaCaTs and HDMECs (Appendix A). However, at the highest seeding ratio of 1:1, GFP-HaCaTs in the co-culture negatively affected the endothelial tubular networks since fewer endothelial tubes were present.

### 3.2. Mesothelial Cells Positively Affected Branch Point Formation

We quantified the specificity of the alignment of GFP-MCs to the endothelial tubes by assessing their percentage alignments. Our analysis on day 7 of co-culture showed that GFP-MCs had aligned at a mean percentage of 69% (Figure 2A). By comparison, GFP-PCs showed a mean percentage alignment of 44% and GFP-HaCaTs of 21% (Figure 2B,C). Interestingly, the different seeding ratios of labeled cells with HDMECs revealed that higher numbers of GFP-MCs and GFP-HaCaTs led to statistically significant lower alignment percentages (ratios 1:6–1:1), while the alignment of GFP-PCs to endothelial tubes was not significantly affected by different seeding ratios.

In order to determine whether GFP-MC alignment had any beneficial or inhibitory effects on the endothelial tube formation process, we assessed the effect of GFP-MC alignment on branch point formation of the endothelial tube network. In angiogenesis control cultures where no third cell type (MCs, PCs, HaCaTs) had been added, a mean of 96.6 branch points (±8.9 SEM) were determined in a 1.29 mm^2^ field of view (FoV). In co-cultures with GFP-MCs, the average number of branch points per FoV increased with higher seeding densities of the GFP-MCs from 84 (1:60) to 149 (1:1), and a seeding ratio of 1:1 resulted in a significantly higher number of branch points compared to that of 1:60 (Figure 2D). By comparison, the average number of branch points counted per FoV in all co-cultures with GFP-PCs ranged from 81 (1:1) to 112 (1:6); a statistical significance was detected for the difference between seeding density 1:1 and 1:6, which we consider biologically irrelevant (Appendix A). In GFP-HaCaT co-cultures the average number of branch points per FoV decreased significantly with an increase in GFP-HaCaT seeding numbers (Appendix A), from 112 (1:60) to 24 (1:1), in line with the disruptive effect on the vessel formation observed (Appendix A).

Next, we measured the effect of GFP-MC alignment on endothelial proliferation and health by quantifying the area occupied by the endothelial tubes at the endpoint of the assay. In control angiogenesis cultures of HDMECs with NHDFs, endothelial tubes covered on average 24% of a 1.29 mm^2^ FoV (±0.9% SEM) (Appendix A). By comparison, in GFP-MC co-cultures, on average 27% of a FoV (±0.7% SEM) was covered by endothelial tubes (Figure 2E), which was not significantly different from the control cultures. At all seeding densities tested, GFP-MC alignment resulted in a similar area covered by endothelial tubes, even though a non-significant trend towards a larger area of endothelial tube coverage in cultures with higher GFP-MC seeding numbers was observed. Co-culture with GFP-PCs resulted in an average of 24.2% (±0.4% SEM) of endothelial tube coverage; at seeding ratios of 1:60 and 1:15, the endothelial cell coverage was significantly lower than at 1:6 (Appendix A). By contrast, co-culture with GFP-HaCaT resulted in a slightly lower coverage of endothelial tubes per FoV with an average of 20.8% (±0.7% (SEM)), and increasing GFP-HaCaT seeding numbers resulted in a significant reduction of the area occupied by endothelial tubes (Appendix A).

Taken together, these findings suggest that mesothelial cells aligned specifically with the developing endothelial tubes, especially at lower seeding ratios, which appeared to favor alignment. Interestingly, the highest seeding ratio of mesothelial cells resulted in the highest numbers of endothelial branch points and a non-significant trend towards larger vessel area. In contrast, pericytes showed less specific alignment and overall slightly lower vessel density, area and branch points of endothelial tubes, while HaCaT cells failed to align, and increasing HaCaT density negatively affected the endothelial tube formation across all parameters assessed.

### 3.3. MC Interacted with and Supported Endothelial Tube Formation

Because we had observed a significant effect of GFP-MC presence on branch point formation, we monitored the behavior of GFP-MCs during the culture in the angiogenesis assay using live cell imaging. HDMECs labeled with dTomato and GFP-MCs were seeded at 1:60 in the angiogenesis assay and images were recorded over >115 h in regular intervals in the CellIQ. These recordings revealed that GFP-MCs were engaged near and at the area where branch points formed, demonstrating interaction with the endothelial cells (Appendix A).

The movie stills also demonstrate that the GFP-MCs become more irregularly shaped from around 78 h after the start of the co-culture, suggesting a shift in cell behavior at around this time (Figure 3).

### 3.4. Mesothelial Cells Upregulated Expression of the Pericyte Marker Ng2 When Co-Cultured in the Angiogenesis Assay

To determine whether the specific behavior of the mesothelial cells in the angiogenesis assay was reflected in changes in gene expression, we performed qRT-PCR for a range of mesothelial, EMT, endothelial and mural markers. We used optimized mouse-specific primers to allow us to measure the level of GFP-MC mRNA transcripts in RNA extracted from individual cultures of GFP-MCs (in FGM, Basal, or Basal with VEGF-A_165_), in co-culture with NHDFs with or without VEGF-A_165_, and in co-culture with NHDFs and HDMECs with or without VEGF-A_165_ (angiogenesis assay), or in mouse cardiac microvascular endothelial cells (MCMECs) from the inbred C57Bl/6 mouse strain, mouse dermal microvascular endothelial cells (MDMECs) from the outbred CD1 mouse strain, or mouse aortic SMCs (MASMCs) as control cell types. Our results revealed that expression of the mesothelial marker Mesothelin (Msln) was slightly upregulated in GFP-MCs when co-cultured with NHDFs and stimulated with or without VEGF-A_165_ (GFP-MC+NHDF Basal; GFP-MC+NHDF VEGF-A_165_) when compared to GFP-MCs grown under basal conditions (GFP-MC Basal, Figure 4A). Importantly, Msln was downregulated in GFP-MCs co-cultured in the angiogenesis assay.

The EMT marker Zeb1 was significantly upregulated by 3.6-fold in GFP-MC cultured with NHDF and VEGF-A_165_ compared to GFP-MC Basal (Figure 4B). Expression levels of Zeb1 were at similar levels in GFP-MCs in the angiogenesis conditions (co-culture with HDMECs, NHDFs and VEGF-A_165_) compared to the basal condition on day 7 of co-culture. These results suggest that the mesothelial cells responded specifically to the NHDF-VEGF-A_165_ stimulation, indicating their changes to a more migratory phenotype in line with a role for Zeb1 in the regulation of cell migration [36], while the angiogenesis co-culture conditions at this stage had no pro-migratory effect.

Next, we interrogated expression changes of the pericyte/vSMC markers Pdgfrb and Neuron-glial antigen 2 (Ng2), also known as chondroitin sulfate proteoglycan 4 (Cspg4) [37], under these conditions. Both Pdgfrb and Ng2 were significantly higher expressed in MASMCs than in GFP-MC Basal (Figure 4C,D). GFP-MCs responded to the NHDF-VEGF-A_165_ co-culture by significant 4-fold upregulation of Pdgfrb expression, while being cultured in the angiogenesis assay with or without VEGF-A_165_ led to a non-significant increase (1.9 fold with; 2.4 fold without VEGF-A_165_) in expression compared to GFP-MC Basal (Figure 4C). By contrast, the GFP-MCs significantly upregulated expression of Ng2 by between 4- and 6-fold when co-cultured with NHDFs with or without VEGF-A_165_, or in the angiogenesis assay without VEGF-A_165_ compared to GFP-MC Basal (Figure 4D). While the 3-fold upregulation of Ng2 expression in the angiogenesis assay with VEGF-A_165_ was not statistically significant when compared to GFP-MC Basal, there was also no statistical significance to the expression level of the GFP-MCs in the angiogenesis assay without VEGF-A_165_. These data suggest that mesothelial cells are stimulated by the angiogenesis assay to upregulate expression of the pericyte marker Ng2 and that the co-culture with NDHFs in the presence of VEGF-A_165_ upregulated Pdgfrb expression in mesothelial cells.

Next, we analyzed the expression levels of the smooth muscle cell marker α-smooth muscle actin (SMA) to determine if cells had started to differentiate into vSMC as suggested by their behavior in the angiogenesis assay. Our results showed that SMA expression was very high at the mRNA level throughout the conditions except in MCMECs and MDMECs (Appendix A). These results were unexpected as they suggest that GFP-MCs expressed high levels of SMA under any condition at similar levels to those in MASMCs.

We also determined the expression of CD31 and VE-Cadherin in the GFP-MCs under the described conditions (Appendix A). Our analysis showed that expression for both endothelial markers was exclusively high in the endothelial cell lines, demonstrating that GFP-MCs do not become endothelial cells.

Taken together, our data show that GFP-MCs were stimulated to upregulate the pericyte marker Ng2 in the angiogenesis assay in the presence of endothelial cells. Furthermore, we determined that mesothelial cells exhibited changes towards more migratory phenotype when grown on NHDF feeders and stimulated with VEGF-A_165_, as determined by upregulation of Zeb1 expression. These results suggest a two-step activation process of MCs towards a pericyte marker phenotype, where mesothelial cells co-cultured with the fibroblasts are stimulated to become migratory. However, at the day 7 time point in the angiogenesis assay when endothelial tubes have formed and mesothelial cells aligned, expression of the migratory markers is downregulated, and instead, the pericyte marker Ng2 is upregulated.

### 3.5. Mesothelial Cells Changed Cell Shape in Response to Culture on Fibroblast Feeder Cells and with VEGF

Our findings suggested that the GFP-MCs were stimulated to a migratory behavior by the NHDFs and to perivascular phenotype during co-culture in the angiogenesis assay (Figure 4). In addition, our data showed that the GFP-MCs were subject to cell changes in the angiogenesis assay (Figure 1E and Figure 3). We reasoned that the changes in cell shape could be caused through stimulation originating from the NHDFs and/or VEGF-A_165_.

To address this question, we measured GFP-MC cell shape in response to co-culture with NHDFs and stimulated with VEGF-A_165_. Mesothelial cell shape was not visibly altered when GFP-MCs were mono-cultured in the presence of VEGF-A_165_ compared to basal conditions (Figure 5A,B). By contrast, seeding GFP-MCs on NHDF as feeder cells led to a clear change in cell shape, which was further influenced by the presence of VEGF-A_165_ in the culture medium (Figure 5C,D). We quantified these cell shape changes using Cellpose image analysis software, which revealed statistically significant differences in mean area, mean perimeter and mean circularity between all conditions except between co-culture of GFP-MCs on NHDF with or without VEGF-A_165_ (Figure 5E–G): There was a statistically larger mean cell area in GFP-MCs stimulated with VEGF-A_165_ than at basal condition. The mean cell area was further statistically increased in GFP-MCs cultured on NHDFs with or without VEGF-A_165_ over mono-cultured cells (Figure 5E, Appendix A). Similarly, the mean cell perimeter of GFP-MCs was significantly larger in cells exposed to VEGF-A_165_ compared to basal, and even further increased when cells had been co-cultured with NHDFs in the presence or absence of VEGF-A_165_ (Figure 5F). The circularity of GFP-MCs was significantly lower when stimulated with VEGF-A_165_ and even further reduced when co-cultured with NHDFs with or without VEGF-A_165_ stimulation (Figure 5G). These results indicate that mesothelial cells underwent considerable changes towards more elongated and larger cells, stimulated by both VEGF-A_165_ and co-culture with NHDF, and in the absence of endothelial tube formation. Together, these results suggest that the contact with the NHDFs ‘prepares’ the GFP-MCs for their interaction with the HDMECs in the following days of co-culture.

### 3.6. Mesothelial Cells Showed Migratory Behaviour in Response to Supernatant from HDMEC–NHDF Co-Culture but Not VEGF

Because we had observed a change in cell shape in the GFP-MCs in response to co-culture with NHDFs and VEGF-A_165_, we explored the effect of the supernatant from the co-culture of HDMECs and NHDFs on mesothelial cell migration. In a Boyden chamber assay, we found that there was a 34-fold increase in GFP-MC migration over serum-free medium in basal supernatant from HDMEC–NHDF co-culture on day 4 or day 7 (Figure 6A). By comparison, there was a 21-fold or 27-fold increase in GFP-MC migration over serum-free medium in supernatant from HDMEC–NHDF co-culture containing VEGF-A_165_ on day 4 or day 7, respectively. When serum-free medium was supplemented with VEGF-A_165_, this failed to elicit a strong migratory response compared to full-growth media (Figure 6B).

These results suggest that the mesothelial cells have the potential to respond with significantly increased migratory behavior to one or several secreted, soluble factors in the supernatant of the angiogenesis assay and this response is not VEGF-A_165_-mediated.

### 3.7. Mesothelial Cells Aligned and Supported Tube Formation in a Cardiac Angiogenesis Assay

The observation that mesothelial cells can engage with the human dermal microvascular endothelial cells in the angiogenesis assay during the junction forming process, express SMA and Ng2 and change their cell shape in response to the stimuli (NHDF feeder, VEGF-A_165_) led us to explore their interaction with human cardiac microvascular endothelial cells (HCMECs) cultured on a layer of human cardiac fibroblasts (HCFs) in a ‘cardiac’ angiogenesis assay. When seeding GFP-MCs together with HCMECs on HCFs (experimental design replicated as outlined in Figure 1A), GFP-MCs had interacted and aligned with the endothelial tubes at the experimental endpoint at day 6 (Figure 7A–C). We quantified the effect of GFP-MC alignment in the cardiac angiogenesis assay and observed that adding GFP-MCs to the cardiac angiogenesis assay in the presence of VEGF-A_165_ resulted in a similar-sized vessel area compared to the co-culture of HCMECs on HCFs alone in the presence of VEGF-A_165_ (Figure 7D). Furthermore, the number of branch points in the cardiac angiogenesis assay was similar to that in the cardiac co-culture assay (Figure 7E). Of note, culturing the GFP-MCs with HCMECs on HCFs in the absence of VEGF-A_165_ could not compensate for the angiogenic stimulus that VEGF-A_165_ represents.

### 3.8. Mesothelial Cells Integrated in a Cardiac 3D Spheroid Model

Since GFP-MCs aligned with HCMECs in the 2D cardiac angiogenesis assay and had shown supportive behavior in the angiogenesis assays using the HDMECs, we explored whether mesothelial cells could have a supportive, perivascular role in a more physiologically relevant cardiac cell model. We have previously generated a 3D multicellular human cardiac microtissue composed of cardiomyocytes, cardiac fibroblasts and cardiac microvascular endothelial cells [30]. We utilized the AC16 human cardiomyocyte cell line [38] to generate a multicellular cardiac microtissue with HCFs, HCMECs and GFP-MC cells. Immunofluorescence analysis after 7 days of culture revealed that the GFP-MCs had integrated into the spheroids and aligned with the HCMECs which formed a rudimentary endothelial network (Figure 8).

These data indicate that under optimal conditions, mesothelial cells integrate well into cardiac spheroids where they provide support and alignment for the HCMEC network.

## 4. Discussion

Mesothelial cells are a unique type of epithelial cells that line the serous cavities, including the pleural, pericardial and peritoneal cavities. They serve as a protective barrier, provide lubrication and facilitate organ movement within these cavities. Mesothelial cells have been shown to have angiogenic properties in in vivo studies [6] and to contribute to the development of vascular smooth muscle cells [5]. However, the contribution of MCs to microvascular structure remains obscure. In this study, we used variations of co-culture in vitro angiogenesis assays to analyze the potential of murine MCs to affect microvascular behavior. Our results revealed that MCs undergo transcriptional and morphological changes on contact with fibroblasts ultimately allowing migration and close association with endothelial cells.

We have previously established and characterized the mouse mesothelial cells that we derived from mouse omentum primary cultures and transduced with a GFP-lentivirus [24]. The in vitro angiogenesis assay using human dermal fibroblasts and human dermal microvascular endothelial cells had been previously established in our lab [22]. Here, we co-cultured the GFP-MCs with the human cells in the angiogenesis assay to explore the interaction between mesothelial and endothelial cells. Equivalent human mesothelial cells are not readily available and immortalized human mesothelial cells may not display a primary mesothelial phenotype. Furthermore, the use of the mouse mesothelial cells allowed us to specifically dissect out the changes in gene expression in the mesothelial cells by using mouse-specific primers in the qRT-PCR analysis. Future studies would benefit from using well-characterized primary human mesothelial cells; however, the process of lentivirus transduction and expansion of the human primary cells may exhaust their proliferation potential and primary characteristics.

The in vitro angiogenesis assay revealed that GFP-MCs have the ability to align with endothelial cells, and specifically associate at endothelial branch points. Our spatial and temporal analysis suggests that this interaction promoted an increased number of branch points and may have also affected the total vessel area. The observed co-localization of GFP-MCs with endothelial cells was reminiscent of pericyte behavior in vivo, where intravital imaging in mice has revealed that pericytes associate with endothelial tip cells at branch points [39].

The acquisition of a perivascular/mural phenotype by GFP-MCs was further supported when we performed species-specific qRT-PCR analysis of the pericyte marker *Ng2* [37] in RNA extracted from the angiogenic co-cultures comprising GFP-MCs and/or fibroblasts and/or endothelial cells. Murine *Ng2* expression significantly increased when the GFP-MCs were plated on fibroblasts or fibroblasts together with endothelial cells. This suggests that contact of GFP-MCs with fibroblasts was critical in promoting a more perivascular phenotype.

Mesothelial cells have been shown to undergo mesothelial–mesenchymal transition (MMT) and acquire migratory and invasive properties [40]. Analysis of murine *Zeb1* expression, a marker of EMT [41], revealed increased expression of *Zeb1* when GFP-MCs were plated on fibroblasts. Interestingly, plating the GFP-MCs on fibroblasts and endothelial cells did not show a concomitant increase in *Zeb1* expression as observed when GFP-MCs were cultured on the fibroblasts alone, suggesting that longer-term exposure to endothelial cells may ultimately suppress *Zeb1* expression in the GFP-MCs.

Consistent with the acquisition of a migratory behavior, analysis of cell shape revealed that GFP-MCs plated on fibroblasts or fibroblasts together with endothelial cells, underwent a shape change, increasing cell area and perimeter and reducing circularity. This more elongated morphology allows for the formation of leading edges and trailing tails, facilitating directional migration [42]. Our findings suggest that contact of the GFP-MCs with fibroblasts appears to prime this increased migratory behavior, potentially through MMT. Whilst cell–cell contact could drive this process, the ability of conditioned media from the fibroblasts and endothelial cells to stimulate GFP-MC migration directly suggested that a soluble secreted factor/cytokine, possibly from the fibroblasts, was responsible for driving the migratory behavior in the GFP-MCs. Our current research is aimed at trying to identify the potential soluble factors that can influence GFP-MC migration and potentially MMT.

Mesothelial cells have been shown to play an important role in serosal homeostasis and repair following injury [2]. Establishing and supporting new vasculature is critical for tissue regeneration. Genetic fate-mapping of epicardial cells in zebrafish suggests that these cells may contribute to cardiac repair by forming peri-vascular cells [43,44]. Loss of pericyte function has been implicated in pathophysiological conditions such as drug-induced cardiotoxicity [45], diabetic retinopathy and ischemia [46]. Analysis of the behavior of the GFP-MCs in a 3D cardiac microtissue composed of cardiac myocytes, cardiac endothelial cells and cardiac fibroblasts revealed that the GFP-MCs were able to closely associate with the cardiac endothelial cells in this physiologically relevant model. These 3D cardiac microtissues represent important physiologically relevant in vitro models used for pre-clinical drug safety testing [30] and for analyzing cardiac regeneration [47] and will allow further interrogation of the ability of mesothelial cells to undergo MMT to perivascular cells and support vascularization.

## 5. Conclusions

This study suggests that mesothelial cells have the potential to adopt a mural phenotype by aligning closely and specifically with endothelial networks in vitro, upregulating the pericyte marker *Ng2*. In addition, stimulated mesothelial cells exhibited characteristics of migratory cells by changing their appearance towards less circular cells and expressing markers involved in the regulation of EMT. Finally, mesothelial cells showed alignment with cardiac endothelial cells in 3D cardiac spheroids, suggesting their potential as perivascular cells.

## Figures and Tables

**Figure 1 cells-12-02436-f001:**
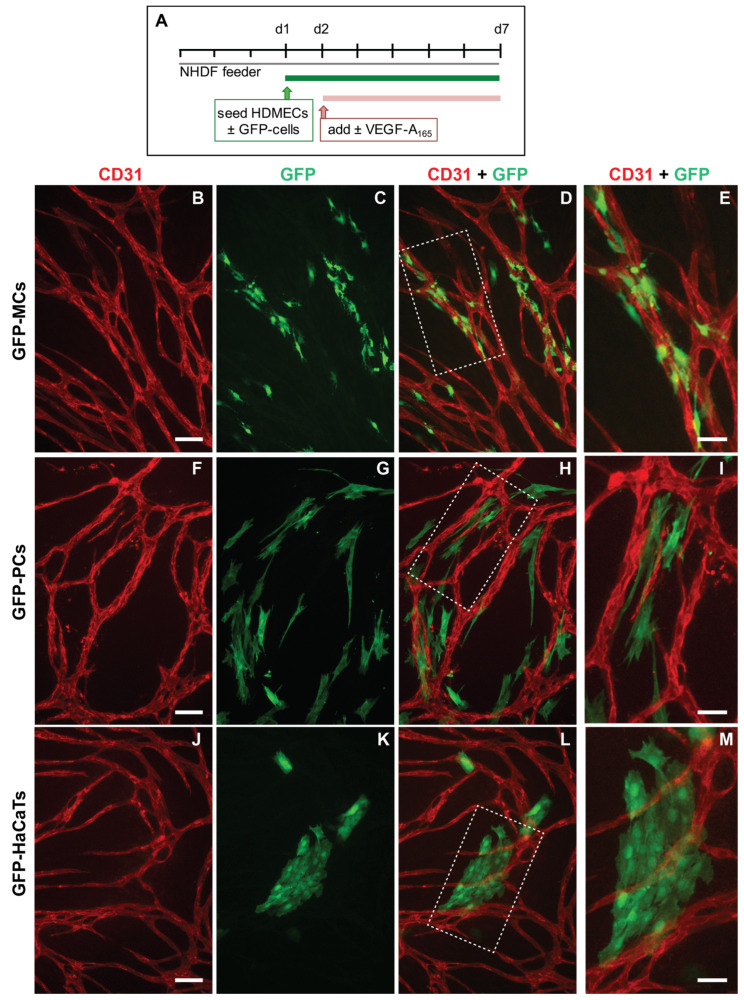
Alignment of mesothelial cells (GFP-MCs) to endothelial tubes (HDMECs) in vitro in the presence of VEGF-A_165_. (**A**). Experimental design. (**B**–**E**). Alignment of mesothelial cells to HDMEC tubular networks at a seeding ratio of 1:15 (2000 GFP-MCs:30,000 HDMECs). Arrowheads point towards close interaction between endothelial tubes and mesothelial cells with extended cell projections (**E**). (**F**–**I**). Alignment of human pericytes to HDMEC tubular networks at a seeding ratio of 1:15 (2000 GFP-PCs:30,000 HDMECs). (**J**–**M**). Arrangement of HaCaTs with HDMEC tubular networks at a seeding ratio of 1:15 (2000 GFP-HaCaTs:30,000 HDMECs). Scale bars (**B**–**D**,**F**–**H**,**J**–**L**) are 100 μm and (**E**,**I**,**M**) are 50 μm.

**Figure 2 cells-12-02436-f002:**
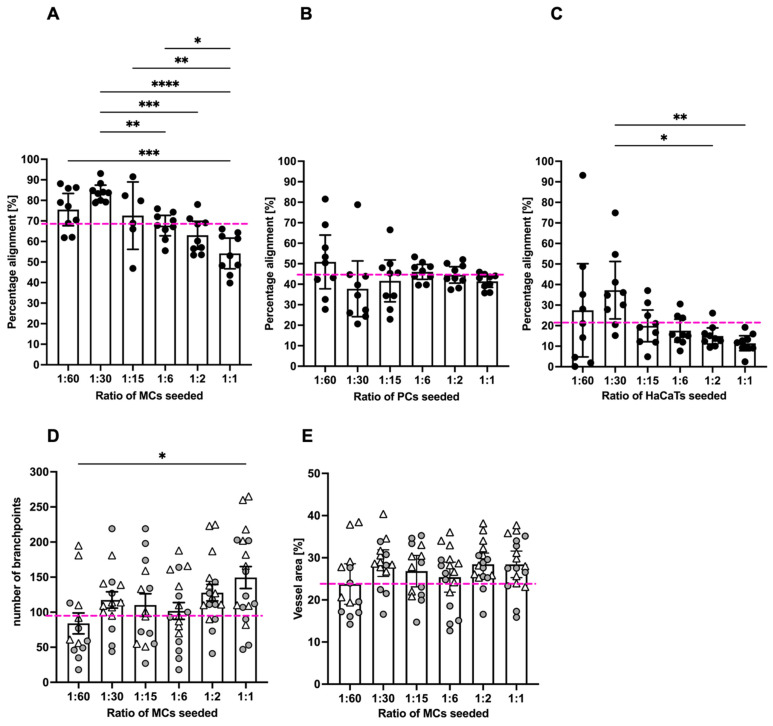
Quantification of mesothelial cell alignment and the effect on endothelial tube formation after 6 days of co-culture. (**A**). GFP-MCs seeded at ratios of 1:1 to 1:60 with HDMECs showed on average 69% alignment across all seeding densities. Significant differences in percentage alignment were observed amongst different conditions, with the lowest alignment at the highest GFP-MC seeding density. (**B**). GFP-PCs seeded at ratios of 1:1 to 1:60 with HDMECs showed on average 44% alignment across all seeding densities. There were no significant differences in percentage alignment across the conditions. (**C**). GFP-HaCaTs seeded at ratios of 1:1 to 1:60 with HDMECs showed on average 21% alignment across all seeding densities. There were significant differences in percentage alignment between 1:30 and 1:1 or 1:2 seeding ratios. Generally, a higher seeding density led to a lower alignment. (**D**,**E**). The presence of GFP-MCs in the co-cultures led to an increase in branch points with increasing GFP-MCs seeded (**D**), and a slightly larger vessel area in the endothelial network, but without significant difference. In (**A**–**C**), the pink stippled line indicates the average values across each condition; and in (**D**,**E**), the pink stippled line demarcates the control values for branch points and percentage vessel area in the absence of GFP-MCs. Circles and triangles indicate data points from two independent experiments. Data were analyzed by one-way ANOVA with Tukey’s multiple comparison test; data are shown as mean with a 95% confidence interval; significance was defined by a *p*-value ≤ 0.05. *p*-value ≤ 0.05 (*), <0.005 (**), ≤0.0005 (***), ≤0.0001 (****).

**Figure 3 cells-12-02436-f003:**
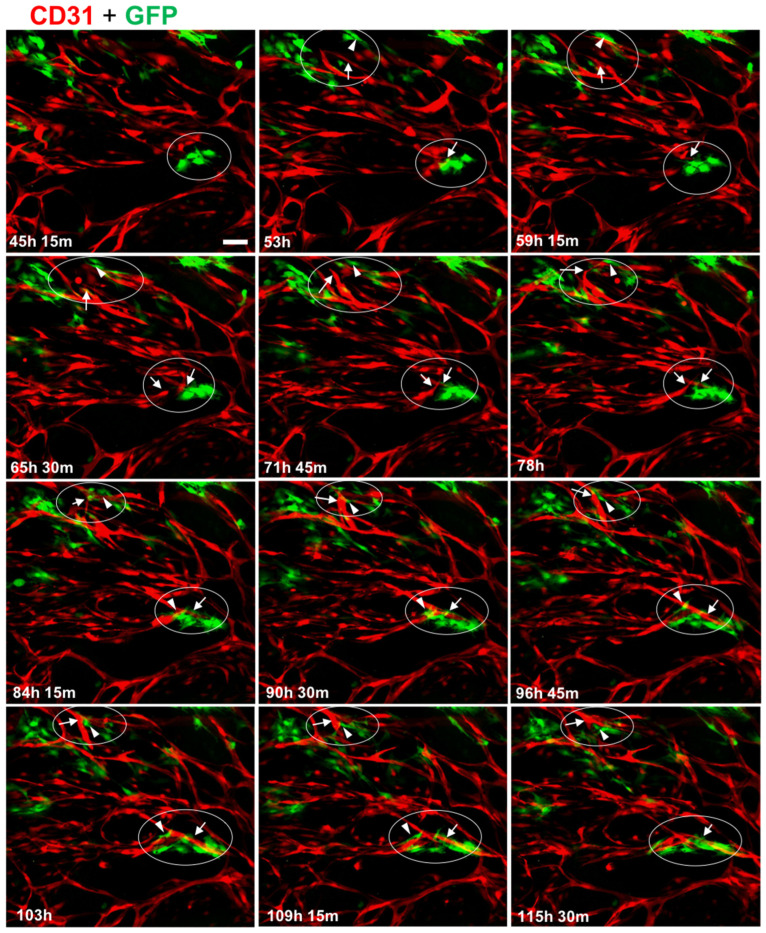
Mesothelial cells interact with endothelial tube formation process at junction points. Collection of individual images captured over a 70 h period, starting from day 2 (45 h, 15 min) of the co-culture. In two areas (encircled), GFP-MC (green, GFP) interaction with branching processes by the HDMECs (red, CD31) leading to junctions in the tubular network are shown. Arrows point toward endothelial-junction-forming process and arrowheads point to GFP-MCs that appear to engage actively by changing their shape into elongated migratory cells. Scale bar 100 μm.

**Figure 4 cells-12-02436-f004:**
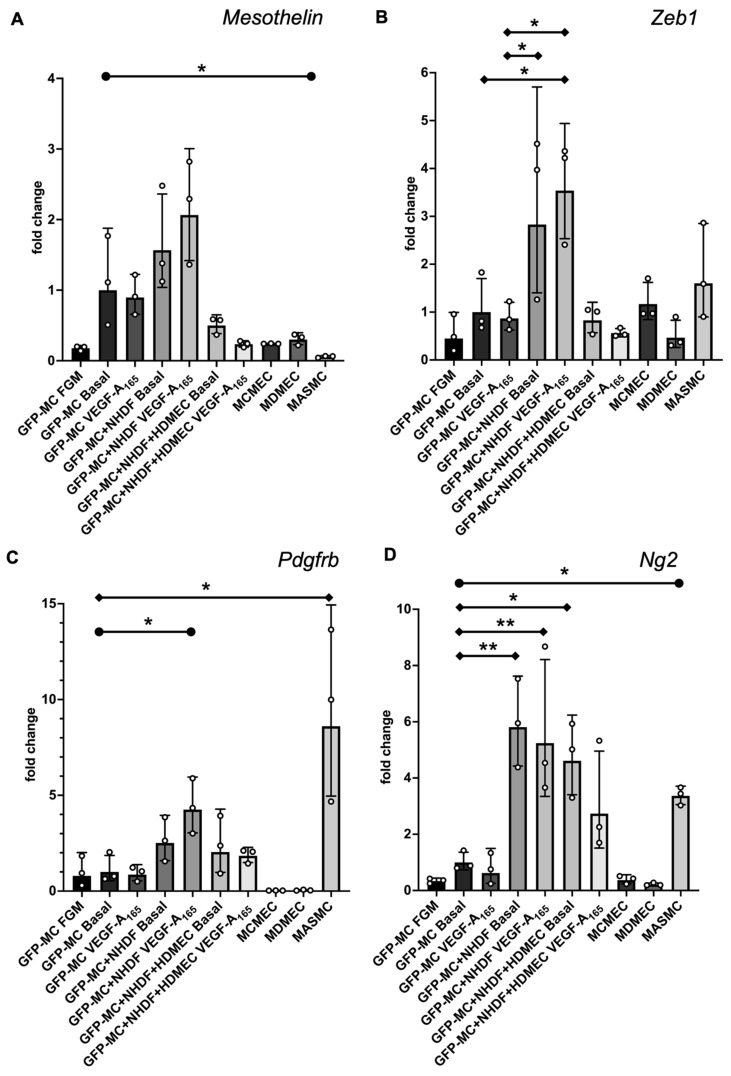
Changes in gene expression of GFP-MCs in response to VEGF-A_165_ stimulation, co-culture with NHDF, or in the angiogenesis assay after 6 days of culture by qRT-PCR. (**A**). There was no statistically significant difference in Mesothelin expression for any combination when compared to GFP-MC Basal using a one-way ANOVA, although a trend to 2-fold higher expression in GFP-MC+NHDF stimulated with VEGF-A_165_ compared to GFP-MC Basal could be observed. There was a significantly lower expression of *Mesothelin* in mouse primary arterial smooth muscle cells (MASMCs) compared to GFP-MC basal using an unpaired *t*-test with *p* = 0.0421. (**B**). There was a statistically significant 3.6-fold higher *Zeb1* expression for GFP-MC+NHDF stimulated with VEGF-A_165_ compared to GFP-MC Basal or GFP-MC with VEGF-A_165_, and a 2.8-fold higher expression in GFP-MC+NHDF without VEGF-A_165_ compared to GFP-MC Basal using a one-way ANOVA with Tukey’s multiple comparison test. (**C**). Expression analysis for *Pdgfrb* revealed a statistically significant 8-fold higher expression in MASMCs compared to GFP-MC Basal using a one-way ANOVA with Tukey’s multiple comparison test. *Pdgfrb* expression was 4-fold higher in GFP-MC+NHDF with VEGF-A_165_ compared to GFP-MC Basal; this was significant when using an unpaired *t*-test. (**D**). *Ng2* was significantly higher expressed in GFP-MC+NHDF without VEGF-A_165_ (5.9-fold), in GFP-MC+NHDF stimulated with VEGF-A_165_ (5.4-fold) and in the GFP-MCs in angiogenesis assay without VEGF-A_165_ (4.7-fold), compared to GFP-MC Basal using a one-way ANOVA with Tukey’s multiple comparison test. There was a 3.4-fold significantly higher expression in MASMCs when compared to GFP-MC Basal by unpaired *t*-test. Data displayed as geometric mean with geometric SD, significance was defined by a *p*-value ≤ 0.05. *p*-value ≤ 0.05 (*), ≤0.005 (**). FGM—full growth media, MCMEC—mouse primary cardiac microvascular endothelial cells, MDMEC—mouse dermal microvascular endothelial cells.

**Figure 5 cells-12-02436-f005:**
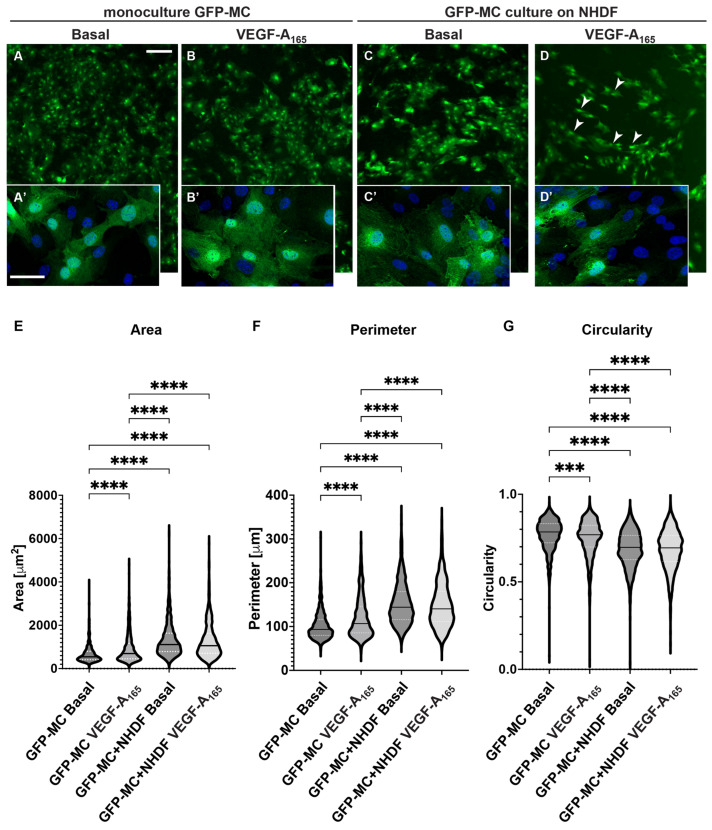
Changes in cell shape of GFP-MCs in response to VEGF-A_165_ and culture on an NHDF feeder layer. (**A**–**D**). GFP-MCs were cultured on plastic (monoculture, (**A**,**B**)) or NHDFs (**C**,**D**) for 6 days followed by immunofluorescence staining for GFP and DAPI nuclear stain. Scale bars for (**A**–**D**) 200 μm; for (**A’**–**D’**) 50 μm. (**E**–**G**). GFP-stained MCs under the four culture conditions were assessed using the Cellpose workflow and Fiji for area, perimeter and circularity. A Kruskal–Wallis test with Dunn’s multiple comparison tests determined significant differences across the four conditions for the three parameters; data shown as violin plots with median and quartiles. Significance was defined by a *p*-value ≤ 0.05. *p*-value ≤ 0.0005 (***), ≤0.0001 (****).

**Figure 6 cells-12-02436-f006:**
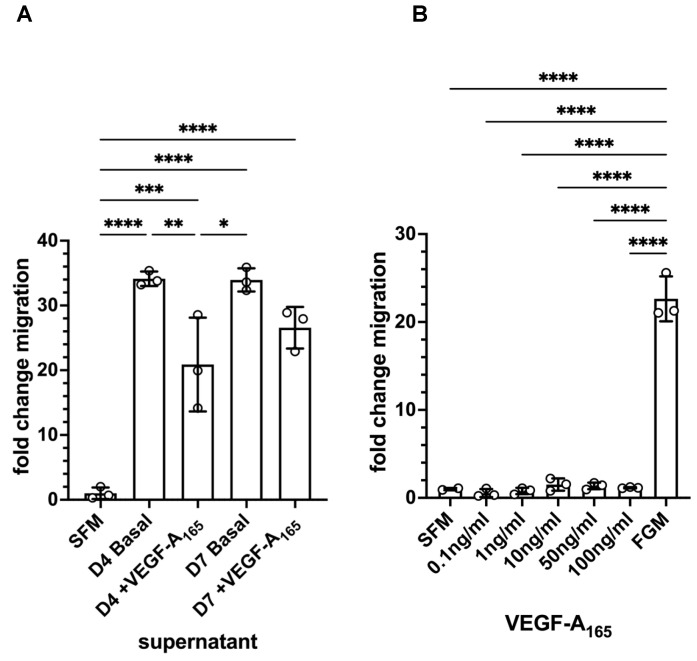
Quantification of GFP-MC migratory behavior in response to supernatant and VEGF-A_165_ in a Boyden chamber assay. (**A**). GFP-MCs were cultured for 16 h in a Boyden chamber exposed to serum-free medium (SFM), or supernatant collected on day 4 from HDMEC–NHDF angiogenesis co-culture with basal media (no VEGF-A_165_) (D4 Basal), or with full angiogenesis media (with VEGF-A_165_) (D4 +VEGF-A_165_), or supernatant collected on day 7 from angiogenesis cultures with basal media (D7 Basal), or full angiogenesis media (D7 +VEGF-A_165_). (**B**). The exposure of GFP-MCs to a range of concentrations of VEGF-A_165_ over 16 h failed to lead to migratory behavior compared to SFM, and full growth media (FGM). Statistical analysis by one-way ANOVA with Tukey’s multiple comparisons posthoc test. Data displayed as mean with standard deviation. Significance was defined by a *p*-value ≤ 0.05. *p*-value ≤ 0.05 (*), ≤0.005 (**), ≤0.0005 (***), ≤0.0001 (****).

**Figure 7 cells-12-02436-f007:**
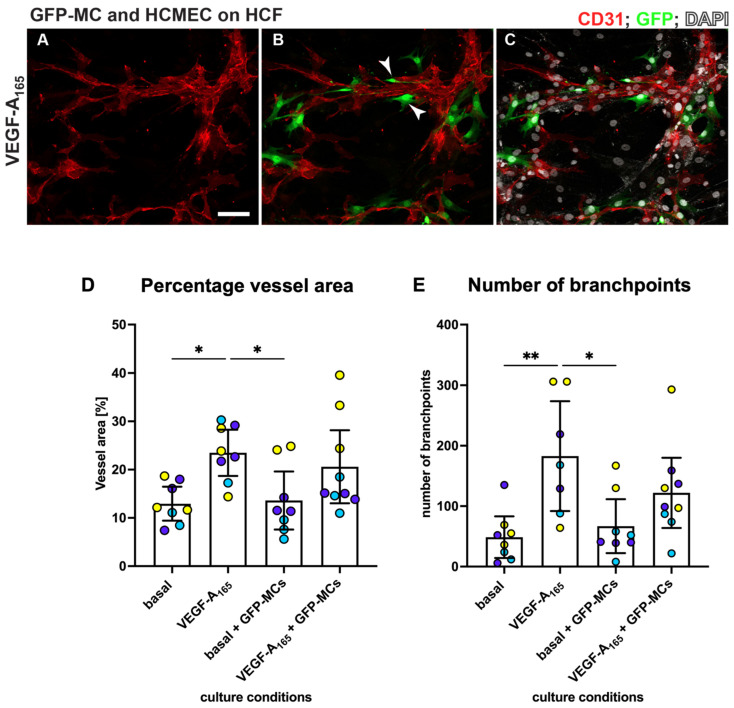
MCs align with HCMECs in angiogenesis co-culture. HCMECs (CD31, red) were cultured on HCF with GFP-MCs (GFP, green) in medium containing VEGF-A_165_. Nuclei were labeled with DAPI and are shown in white. Addition of GFP-MCs (ratio 1:60) to HCMEC-angiogenesis cultures in medium with VEGF-A_165_ (**A**–**C**) revealed alignment of the MCs to the tubes. Arrowheads in (**B**) point towards GFP-MCs that align with HCMECs. (**D**,**E**). Quantification of angiogenesis of HCMECs on HCFs and with/without GFP-MCs. Percentage vessel area and number of branch points are highest when HCMECs are cultured with VEGF-A_165_. GFP-MCs added to the co-culture resulted in similar values (no statistical significance). Yellow, purple and turquoise represent data points from 3 independent experiments, and data are shown with a 95% confidence interval. Scale bar 100 μm. Statistical analysis by one-way ANOVA with Tukey’s multiple comparisons test. Images representative of three independent experiments. Significance was defined by a *p*-value ≤ 0.05. *p*-value ≤ 0.05 (*), ≤0.005 (**).

**Figure 8 cells-12-02436-f008:**
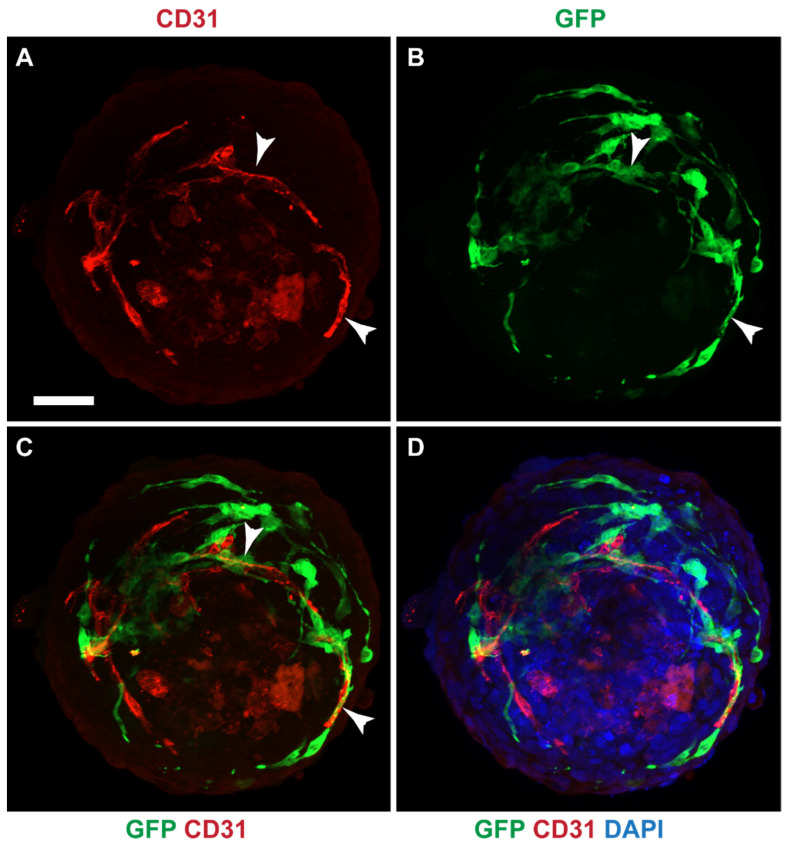
GFP-MCs aligned with HCMECs in a cardiac spheroid model. Immunofluorescence labeling for GFP (green), CD31 (red) and nuclear stain (DAPI, blue). (**A**–**D**). GFP-positive mesothelial cells were detected closely aligned with endothelial tubes (arrowheads). Scale 50 μm. Data are presented from one experiment.

## Data Availability

All datasets from this study will be made available on request.

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
