# Peer review of "Mesothelial Cells Exhibit Characteristics of Perivascular Cells in an In Vitro Angiogenesis Assay"

_cells, 2023, doi:10.3390/cells12202436_

Round 1

Reviewer 1 Report

In this manuscript, the authors describe a multitude of experiments, leading to the conclusion that mesothelial cells can adopt pericyte features in contact with endothelial cells.

The manuscript is written well and most of the experiments are performed adequately.

However, there is one major point preventing publication in the present state:

1.) Do the MCs really cover BECc or rather make contact to LECs?

Regarding the HDMECS purchased from Promocell: These cells are at least to my experience not only blood microvascular EC, but comprise a significant amount of lymphatic microvascular EC. I guess Promocell also knows this, as they also offer dedicated blood- or lymphatic microvascular ECs.

CD31 marks both of them. It may be of crucial interest to know whether MC prefer the one or the other. This may also help to understand the differences between MC and real PC in Fig. 2, Suppl. Figs. 3+5. Especially the fact that MC have direct connections to lymphatics, leading to open lymphatics towards the peritoneal cavity (called “Stomata”). This close relation may influence the interpretation of your results and may lead to different conclusions.

It is therefore necessary to use LEC markers like Prox1, Podoplanin, or Lyve1 to determine cell identities together with MC adhesion.

The same may apply also to cardiac microvascular EC or any of the mouse EC used. LECs are often co-enriched during isolation processes.

Minor Points:

1.) Regarding your real-time PCR experiments, it remains unclear how Ct-values obtained were processed in further calculation leading to the given values in Fig. 4. Please explain your calculation.

2.) Suppl. Fig. 4: E) + F) should rather be A) + B)?

3.) Fig. 5: Typo in axis titel: Perimenter → Perimeter?

Reviewer 2 Report

This study clarified that mesothelial cells have the ability to acquire a perivascular phenotype and associate with endothelial cells by in vitro angiogenesis assay

This conclusion is based on a sufficient amount of results utilizing the experimental system developed by the authors.

However, the paper needs to be revised due to the following issues.

There are some areas where the fonts are different. All should be unified. (L173-182, L240, L558-L563)

Sup Fig3 and Sup Fig4

 The figures show each other are reversed.

Sup Fig4

 Figure legend does not correspond to figures. where did (E) and (F) it come from?

L296

 1:30->1:60 ?

L337

 Figure 1D->Figure 2D

L338-339 "and the seeding density of 1:1 was significantly lower than 1:6 (Supplementary Figure 7A)."

 While this result is mathematically correct, the results of the statistical analysis here do not show anything new, and the purpose of Sup Fig 7A is only to emphasize the difference between MC and PC. It should be removed to prevent misreading by the reader.

L346-347 "In control angiogenesis cultures of HDMECs with NHDFs, endothelial tubes covered on average 24% of a 1.29mm2 FoV (+/- 0.9% SEM)."

 A picture corresponding to this experiment should be inserted in the supplemantal figure as long as it is being compared as a control.

Figure 2E

 It is very interesting that vessel area showed an increasing trend (although not significant) as high MCs density, even though there are only a few cases where the exclusive area of the endothelial cell tube increased.

 I think your study would be more fascinating if you mentioned this instead of concluding as you did in L349-350 and L360-361... ,,

Figure 4A

 I see a horizontal line on the figure legend, but what is this? I am not at all sure whether the text is correct or the figure is correct.

 The figure should be written the same way as 4B-4D because at first glance it is not clear what gene is being analyzed.

L483

 GFM-MCs->GFP-MCs

Figure 7D and 7E (in graph axis labels)

 GFP-MSCs->GFP-MC(s)

Figure 7E

 What are your thoughts on why GFP-MC was found to increase the branching of HDMECs cultured on NHDF feeder cells (Fig 2D), whereas GFP-MC had no effect on the branching of HCMECs cultured on HCF feeder cells? This should be added in discussion section.

L563 (also L173, L387, and L491)

 MC-GFP->GFP-MC

L614

 epithelial-mesenchymal transition (EMT)->EMT

English readability is not a problem.

Round 2

Reviewer 2 Report

Since sufficient responses to the comments have been made, it is acceptable in present form.